# Mediterranean Diet and Motivation in Sport: A Comparative Study Between University Students from Spain and Romania

**DOI:** 10.3390/nu11010030

**Published:** 2018-12-22

**Authors:** Ramón Chacón-Cuberos, Georgian Badicu, Félix Zurita-Ortega, Manuel Castro-Sánchez

**Affiliations:** 1Department of Education. University of Almería, 04120 Almería, Spain; rchacon@ual.es; 2Department of Physical Education and Special Motility, Faculty of Physical Education and Mountain Sports, Transilvania University of Brasov, 500068 Brasov, Romania; 3Department of Didactics of Musical, Plastic and Corporal Expression, University of Granada, 18071 Granada, Spain; felixzo@ugr.es (F.Z.-O.); manuelcs@ugr.es (M.C.-S.)

**Keywords:** Mediterranean diet, motivational climate, sports, university students

## Abstract

Background: The Mediterranean Diet (MD) is one of the healthiest dietary models worldwide, being an essential mean of preventing pathologies along with the practice of physical activity. Through a comparative study carried out across different countries, it has been demonstrated how this type of habits vary depending on the geographical context. The aim of this research was to evaluate the adherence to MD and its relationships with motivational climate in sport on a sample of university students from Spain and Romania; Methods: A cross-sectional study was conducted on a sample of university students [specialization: Physical Education (*n* = 605; 20.71 ± 2.42 years old)], using as main instruments the Mediterranean Diet Quality Index (KIDMED) for students and adolescents and the Perceived Motivational Climate in Sport Questionnaire-2 (PMCSQ-2); Results: It was shown that students from Spain had a high adherence to the MD (6.65 ± 2.63 vs. 5.06 ± 1.31). Spanish university students got higher scores in task-oriented motivational climate (4.03 ± 0.62 vs. 3.11 ± 0.55) while ego-oriented climate was higher in university students from Romania (3.24 ± 0.54 vs. 2.07 ± 0.75). Finally, it was observed that the task-oriented motivational climate was related to a lower adherence to MD in Spanish students (4.49 ± 0.37 vs. 3.98 ± 0.62). In contrast, in Romanian youth, a medium adherence to the MD was associated with higher scores for the ego-oriented motivational climate (3.27 ± 0.53 vs. 3.00 ± 0.54); Conclusions: As main conclusions, it was shown that the students from Spain had a high adherence to the MD. In addition, it has been demonstrated that ego-oriented climates are linked to a better adherence to MD, especially due to the importance of following a proper diet in sport contexts, as demonstrated by young Romanians.

## 1. Introduction

Being overweight (body mass index of 25 or higher) and its associated pathologies have become a problem in society nowadays, which is why the importance of following a healthy diet and lifestyle has gained much prominence [1,2,3]. The diet models that exist in the world are numerous and varied, each one with particular characteristics [3]. Dietary habits represent important elements for the quality of life and especially for young people [4]. Utilizing different dietary models, several authors state the importance of making a change in people’s eating habits in European countries in order to reduce the risk of different types of diseases [5,6]. The Mediterranean Diet (MD) is one of the most studied model as it is associated with different benefits, such as a better health due to a high consumption of natural antioxidants and a low intake of fats. In addition, MD improves the cognitive function, decreases the risk of diabetes, bone diseases and overweight [6,7,8].

This dietary model is characterized by typical foods of the Mediterranean basin such as legumes, fruits, cereals and olive oil, with a moderate consumption of eggs, fish and dairy products [4,9,10]. It has been shown that people with a good adherence to MD have a high quality of life and life expectancy [11], as this diet has several benefits on public health [12], social relationships [13], sport performance [14] and emotional factors [15,16]. Furthermore, the influence of MD on young people living around the Mediterranean Sea has been studied in countries such as Cyprus, Italy, Creece, Turkey, Spain, Lebanon and Croatia [4,13,15,16,17,18,19,20,21]. Additionally, this dietary model has been analysed in various cultural and socio-economic contexts [10,22,23].

Several studies demonstrate the benefits of a high adherence to the MD [24,25,26,27]. This dietary model helps to improve body composition, favouring the decrease in the percentage of fat mass thanks to its low consumption in hypercaloric foods and foods rich in saturated fats [25]. In addition, MD helps to prevent diabetes, since this model is based on the consumption of foods with a low glycaemic index, such as fruits, vegetables and cereals [26]. Another benefit lies in the Non-alcoholic fatty liver disease, since Trovato et al. [27] demonstrated in an intervention study how a better adherence to MD improved the progress of this pathology due to a greater caloric restriction that generates temporary changes in the liver such as a better sensitivity to hepatic insulin or a lower content of triglycerides. Finally, O’Neil et al. [28] highlight some of the benefits of MD at cognitive level, such as an improvement of states of depression and anxiety or a better academic performance. 

Regular and adequate Physical Activity (PA) in addition to following a healthy diet is essential for the improvement of health. Several studies show how performing a minimum of 90 minutes of moderate PA weekly helps to improve health, body composition or bone mineral density, as well as pulmonary and cardiovascular functions [29,30]. In addition, regular PA decreases anxiety and stress and improves self-concept, well-being and self-esteem [31,32]. For these reasons, promoting this type of habits is essential among university students, since many habits developed in early adulthood are reinforced later in life [21]. In this sense, it is essential to study the motivational aspects that allow to create adherence to the practice of PA, which will be linked to other healthy habits such as adopting a healthy diet.

The Achievement Goals Theory has been one of the most used explanatory models in order to explain the motivational processes involved in sports practice [33,34]. This theory establishes that people set objectives in sports, which will be related to the person’s perception of their own abilities. Therefore, the goals pursued by an individual can be linked to motivational climates oriented toward the task or the ego [33,35]. In the first case, the task-oriented climate will be linked to team work, to higher levels of effort to learn and the assignment of an important role in the task [36]. The ego-oriented climate is related to people whose goals depend on the social recognition that they will achieve through competition within the sport context [36,37]. Considering the premises exposed by Ryan et al. [38] in the Self-Determination Theory, the task-oriented climate will be associated with intrinsic motivations, while the ego climate will be related to extrinsic motivations [39]. In this regard, it has been shown that certain types of motivations can be linked to healthy or non-healthy behaviours [40,41,42]. 

Studies such as those carried out by Chacón et al. [43] or Erturan-Ilker et al. [44] analyse the relationship between motivation in sport and different healthy habits, such as the consumption of harmful substances, the practice of PA or adherence to MD. Although the adherence to MD has been studied in association with physical and healthy lifestyles, a few studies connect it with motivation in sport, as well as with cultural-geography factors. Thus, it is interesting to study these variables in order to promote MD in non-Mediterranean populations [19,45,46]. In view of the above, the present study will provide novel data about the importance of the MD in relation to sociodemographic and motivational parameters in a sample of university students of Physical Education from two countries: Spain and Romania.

The present study establishes two aims: (a) to assess the level of adherence to MD in a sample of university students from Spain and Romania; (b) to analyse the relationship between adherence to MD and motivational climate in sport depending on the country (Spain or Romania); (c) to contrast a structural equation model about the relationship between MD and motivational climate using multi-group analysis.

## 2. Materials and Methods

### 2.1. Subjects and Design

This non-experimental, cross-sectional and descriptive research was conducted on a sample of 651 university students from Spain and Romania, enrolled in the Physical Education degrees, aged between 18 and 24 years old (20.71 ± 2.42 years). The respondents participated voluntarily after receiving a detailed explanation of the objectives and nature of the study. Written informed consent was provided. We excluded from the analysis 46 participants that did not complete the inclusion criteria correctly (i.e., incomplete questionnaires, did not hand informed consent forms), thus the final sample was comprised of 605 subjects (368 males and 237 females): Romania (*n* = 178) and Spain (*n* = 427). The students from Romania were enrolled in the Transylvania University of Brasov, while the Spanish students were enrolled in the University of Granada. Finally, an assumed sampling error of 0.05 was taken into account, considering a random sampling by natural groups [47].

### 2.2. Instruments

Test of Adherence to Mediterranean Diet (KIDMED) [48]. This scale comprises 16 items with an affirmative or negative response; for example: “You eat fruit every day”, which are related to patterns associated with the MD. Twelve of these items have positive connotations (+1) while the other four have negative values (−1). The final score ranges from −4 to +12. This instrument has a reliability of *α* = 0.86.

Perceived Motivational Climate in Sport Questionnaire (PMCSQ-2, [49,50]). This instrument is composed by 33 items which are scored by a five-point Likert-scale, ranging from 1 to 5 (Strongly Disagree-Strongly agree). The scale comprised two factors, higher-order dimensions, each containing three indicators. For task-oriented climate the indicators were: effort/improvement, cooperative learning and important role. For ego-oriented climate the indicators were: unequal recognition, member rivalry and punishment for mistakes. The internal consistency for this questionnaire was *α* = 0.85 for task-climate and *α* = 0.82 for ego-climate.

### 2.3. Procedure

First, we proceeded to request the approval of the study by the Ethics Committee of the University of Granada, which was granted with code "641/CEIH/2018”. Subsequently, the collaboration of educational centres was requested through an informative letter elaborated by the Area of Corporal Expression of the University of Granada. In addition, the informed consent of the respondents was requested through a document in which the nature of the study was detailed.

The data was collected during regular classes in the different university campus. Different research assistants were present in the data collection in order to ensure that questionnaires were properly completed, to provide guidance on the completion of scales and to answer questions. Furthermore, participants receive any incentives.

### 2.4. Data Analysis

Statistical analysis was conducted using the software IBM SPSS® 22.0 (IBM Corp., Armonk, NY, USA). Frequencies and medians were used for basic descriptors, whereas the association between the variables detailed were analysed using T-test and ANOVA-test. In addition, the Kolmogorov–Smirnov’s test was used in order to check the normality of data, as well as the Lillieforts’ correction. Levene’s test was employed in order to check homoscedasticity. Finally, Cronbach’s Alpha coefficient was used to analyse the internal reliability of the instruments used, establishing the Reliability Index at 95%. In addition, AMOS version 23.0 (IBM Corp., Armonk, NY, USA) is used for the structural equation analysis.

The Structural Equation Model (SEM) is composed by seven observed variables and two latent variables (Figure 1). Task-involved climate (TC) and ego-involved climate (EC) represent the exogenous variables, which is inferred by three indicators. Effort/Improvement (EI), Cooperative learning (CL) and Important Role (RI) are the indicators for task-oriented climate while Unequal Recognition (UR), Member Rivalry (MR) and Punishment for Mistakes (PM) are the indicators for ego-oriented climate. On the contrary, the Mediterranean diet (MD) was the observed endogenous variable. The bi-directional arrow (covariance) relates exogenous variables, while the unidirectional arrows are lines of influence between the latent and observable indicators and are interpreted as multivariate regression coefficients. In addition, error prediction terms are associated with endogenous variables. The method of maximum likelihood (ML) was used to estimate relationships between variables. We chose this method because it is consistent, unbiased and invariant to types of scale given variables with a normal distribution.

Model fit was tested using several indices. Chi-squared analysis followed when non-significant p-values indicated good model fit. Normalized Fit Index (NFI), Comparative Fit Index (CFI) and Increase Fit Index (IFI) values higher than 0.90 indicate acceptable model fit while values higher than 0.95 indicate excellent model fit. Root Mean Square Error of Approximation (RMSEA) values below 0.08 indicate acceptable model fit while values below 0.05 indicate excellent model fit.

## 3. Results

Table 1 shows the average scores obtained in each item of the test on adherence to the MD, according to the country of origin of the university students enrolled in Physical Education degrees. In this way, statistically significant differences (*p* ˂ 0.05) were found in fruit and daily fruit juice consumption (0.27 ± 0.44 vs. 0.77 ± 0.41) and in the intake of a daily second fruit (0.78 ± 0.41 vs. 0.42 ± 0.49), obtaining a higher score for Spanish students in the first case and in Romanians in the second. Moreover, Spanish students obtained higher scores for fish consumption (0.42 ± 0.49 vs. 0.67 ± 0.46), pulses more than once a week (0.63 ± 048 vs. 0.71 ± 0.45), cereal consumption (0.71 ± 0.45 vs. 0.86 ± 0.34) and dairy consumption (0.26 ± 0.44 vs. 0.81 ± 0.39), showing statistically significant differences in all the relationships. In addition, students from Romania obtained higher mean values than Spaniards in vegetable consumption (0.88 ± 0.33 vs. 0.33 ± 0.47), pasta and rice consumption (0.76 ± 0.42 vs. 0.42 ± 0.49) and nut use (0.93 ± 0.25 vs. 0.50 ± 0.50). Nevertheless, Romanian students got the highest scores on negative items, showing a higher absolute mean value for fast food (−0.56 ± 0.49 vs. −0.34 ± 0.47), jumping breakfast (−0.46 ± 0.49) vs. −0.31 ± 0.46), processed food intake (−0.89 ± 0.31 vs. −0.12 ± 0.32) and sweets intakes (−0.79 ± 0.40 vs. −0.17 ± 0.37). Finally, Spanish students obtained a higher average value in the total score for adherence to the MD (5.06 ± 1.31 vs. 6.65 ± 2.63).

Scores obtained for motivational climate according to the country are shown in Table 2, showing statistically significant differences at level *p* ˂ 0.001 for all relationships. University students from Spain got higher values for Task-oriented Climate (3.11 ± 0.55 vs. 4.03 ± 0.62) and its categories: Cooperative Learning (2.93 ± 0.75 vs. 4.06 ± 0.72), Effort/Improvement (3.34 ± 0.53 vs. 3.97 ± 0.63) and Important Role (2.87 ± 0.86 vs. 4.10 ± 0.73). Furthermore, students from Romania showed the highest scores in Ego-oriented Climate (3.24 ± 0.24 vs. 2.07 ± 0.75) and its categories: Punishment for Mistakes (3.24 ± 0.61 vs. 1.93 ± 0.79), Unequal Recognition (3.17 ± 0.70 vs. 1.98 ± 0.89) and Member Rivalry (3.39 ± 0.80 vs. 2.56 ± 0.95).

Table 3 shows the relationships between adherence to MD and motivational climate in the university students of both countries. Romanian students revealed statistical differences at level *p* ˂ 0.05 for Ego-oriented Climate (3.27 ± 0.53 vs. 3.00 ± 0.54) and punishment for Mistakes (3.28 ± 0.61 vs. 2.96 ± 0.50) showing higher scores for medium adherence to MD than for higher adherence. In addition, statically significant differences at level *p* ˂ 0.01 were found for Member Rivalry with a high average for medium adherence (3.45 ± 0.76 vs. 2.93 ± 0.96. Moreover, Spanish students showed significant differences at level *p* ˂ 0.05 with higher values for low adherence to MD than medium adherence in Task-oriented Climate (4.49 ± 0.37 vs. 3.98 ± 0.62), Cooperative Learning (4.60 ± 0.37 vs. 4.01 ± 0.72) and Effort/Improvement (4.37 ± 0.53 vs. 3.90 ± 0.65).

Finally, a structural equation model is developed in order to compare the relationships between all the analysed factors according to the country (Figure 2 and Figure 3). Almost all fit indices of model suggested good fit. A significant chi-square value was obtained (*χ*^2^ = 126.85; df = 5.28; *p* ˂ 0.001). Nevertheless, as this index has no upper limit and may also be sensitive to sample size, we also considered other standardized indices which are less sensitive to sample size. The NFI was 0.93, indicating an acceptable fit to the model. The CFI yielded a value of 0.94 and the IFI got a value of 0.94. Finally, RMSEA value was 0.74, indicating acceptable fit.

Table 4 shows the standardized weights of university students from Spain and Romania. A significant negative relationship was found between the two dimensions of motivational climate for university students from both countries, being negative in Spaniards (*p* ˂ 0.001, *b* = −0.429) and positive in Romanians (*p* ˂ 0.001, *b* = 0.872). In relation to the indicators and their association with the exogenous variables, the variable that showed the highest regression weight for the task-oriented climate in Spanish students was the important role (*p* ˂ 0.001, *b* = 0.884), while in the students from Romania it was the effort / improvement (*p* ˂ 0.001, *b* = 0.765). In the case of the ego-oriented climate, the indicator with the highest regression weight was unequal recognition for Spaniards and Romanians (*p* ˂ 0.001, *b* = 0.940; *p* ˂ 0.001; *b* = 0.765). 

Finally, the relationship between adherence to MD and motivational climate in sport is analysed (Table 4). A significant and positive relationship was obtained between the ego-oriented climate and the level of adherence to MD (*p* ˂ 0.05, *b* = 0.103) in young Spaniards, whereas this association was not significant for young Romanians. On the contrary, there was a statistically significant association between the task-oriented climate and the level of adherence to the MD, which was negative (*p* ˂ 0.001, *b* = 0.315). Nevertheless, this relationship was not statistically significant for Spanish university students.

## 4. Discussion

Adherence to the MD has been studied in many countries [4,13,15,16,17,18,19,20,21]. Nevertheless, there are few studies comparing dietary habits and other aspects, such as motivational factors related to the practice of sport. The global score obtained for adherence to the MD was higher for Spanish students. This is due to Romania being a non-coastal country, located further north than Spain. This geographical location, which comes with a different climate and cultural customs, does not enable the cultivation of many of the typical foods of the Mediterranean basin (that constitute the MD) in Romania [6,9,12]. Nevertheless, there is a good amount of food exports in the countries of the European Union, which make these differences less important to the level of adherence to MD, as Baylis et al. [51] show. 

It was found that the consumption of vegetables, fruits and juice is higher in Spanish students than in the Romanian students. This is due to the geographical and cultural characteristics of the Spanish region, as these products are frequently cultivated given the better climatological conditions, an easy access and a low price [12,52]. Furthermore, there is a large production of oranges in Spain. The importance of the intake of this fruit is established in society and juice is one of the most frequent methods of consumption. The benefits of consuming these foods are known since they are rich in vitamins and antioxidants, which encourage their consumption by the Spanish population [52,53].

In addition, it was shown that Romanian students ate fruit or vegetables more than once a day with higher frequency than Spanish students. It has been observed that Spanish students follow a better diet and the consumption of these foods is globalized in general. Nevertheless, they did not achieve much higher scores in the consumption of fruit or vegetables. However, the higher consumption of vegetables and fruit among Romanian students may be justified by the participants who follow a healthy lifestyle. Moreover, the scores can be explained by the anthropometric characteristics of people from Eastern Europe and by the climatic conditions that require a higher food intake. Studies carried out around the world have highlighted the differences that exist between different countries [9,12,54,55,56].

The geographical situation of Spain and its climate favour the consumption of food such as fish, milk or cereals more consumed among its students, whereas the diet model of Romanian students is more oriented towards the consumption of pasta or rice. On the other hand, it is worrisome that the consumption of sweets, processed foods or fast food is higher in Romanian students. This is because they have not adopted the Mediterranean diet within their dietary models and this promotes little or no consumption of these foods [13,52].

The motivational climate in sports practice has been studied in different contexts, mainly the field of Physical Education, competitive sports and healthy habits [36,57,58]. In the present study, higher scores have been observed in the task-oriented motivational climate of Spanish students, whereas Romanian students obtained higher scores in the ego-oriented motivational climate in all its dimensions. This could be explained by the type of sports that are practiced most frequently in these countries. Team sports such as football or basketball are more common in Spain, promoting factors that are more linked to the task climate, such as cooperation or teamwork [50,59]. On the contrary, there is a greater tendency for the practice of individual sports such as wrestling, sports gymnastics or weightlifting in Romania, making them stand out in a greater competitive context with the disappearance of the hedonistic factor [40,60,61]. Therefore, although both sports may link all kinds of motivations, those that involve greater social interaction and collaborative work will be related to more self-determined motivations. This idea, along with the cultural influences of each country, will justify these findings [33,34,60].

In the relationship between the motivational climate and MD, those students who are oriented towards the task are those who have a lower adherence to MD. This can be explained by students that are intrinsically predisposed to the pleasure of physical activity and, thus, do not follow a diet of quality in order to improve their sport performance [41,62]. Nevertheless, the students who got higher scores for ego-oriented climate were those with a medium or high adherence to MD, may propitiated by the importance that has the feeding and the nutrition for the sports performance [6,63]. In addition, the scores in ego-oriented climate showed an increase in relation to medium adherence to MD in Romanian students, which is produced because they do not have low adherence in any case and the importance given to get good results in sport context [41]. On the contrary, the statistical differences are shown in task-oriented climate in Spanish university students. In this students an increase of the average scores of the task climate was linked to a low adherence to the MD. This can be justified by the intrinsic motivations of this group, who do not prioritize nutrition as much as they should, as they do not aim at a better performance like their competitors [64,65,66,67,68]. 

First, the limitations found in this study are determined by the cross-sectional design of this study, where data is collected at a specific time, not allowing to establish causal explanation of this variables. Another limitation is associated with the sample size, since data cannot be generalized to the general population of these countries. Only students between 18 and 24 years of age, enrolled in Physical Education and who have knowledge of nutrition have been analysed. It would also have been interesting to apply the PREDIMED questionnaire but this test that analyses the adherence to the MD in people older than 21 years was not appropriate for our participants (who were mostly less than that age). The variables selected represent another limitation since it would have been of interest to include the socio-economic level and the place of residence of the students in order to know their influence on the level of adherence to the MD. Lastly, with KIDMED being a self-report questionnaire, some comprehension errors could be found, although this is not impeding its application [41,42,43,44,45,46,47,48]. 

## 5. Conclusions

The present study show evidence for a higher adherence to MD among Spanish students, as indicated by a higher intake of fruit, vegetables, cereals and fish. On the contrary, Romanian students ate more pasta, rice and processed products such as sweets or fried foods. In relation to the motivational climate, it was observed that students from Spain obtained higher scores in task climate and students from Romania achieved higher scores in ego climate, due to the type of sports practiced. The relationship between these two variables suggested that the task-oriented motivational climate was related to a diet of poorer quality in young Spaniards, observing the opposite in students from Romania. Finally, the structural equation model revealed how the task-oriented climate exerted a negative influence on the adherence to DM in Romanian students, while the ego-oriented climate exerted a positive influence on the quality of the diet of Spanish students.

It has been shown that the existence of extrinsic motivations can be linked to better nutrition, both for health purposes and the achievement of better sports performance. Although the motivations of intrinsic type are linked to healthier habits associated with sport practice, these were not related to nutrition. While further research is necessary to substantiate the findings from this study, there are adequate indications to pursue development of intervention programs aimed at improving the quality of diet, especially among students in Romania. On the other hand, such programs are also of vital importance for Spanish students, in order to obtain a better quality of the diet in those intrinsically motivated.

## Figures and Tables

**Figure 1 nutrients-11-00030-f001:**
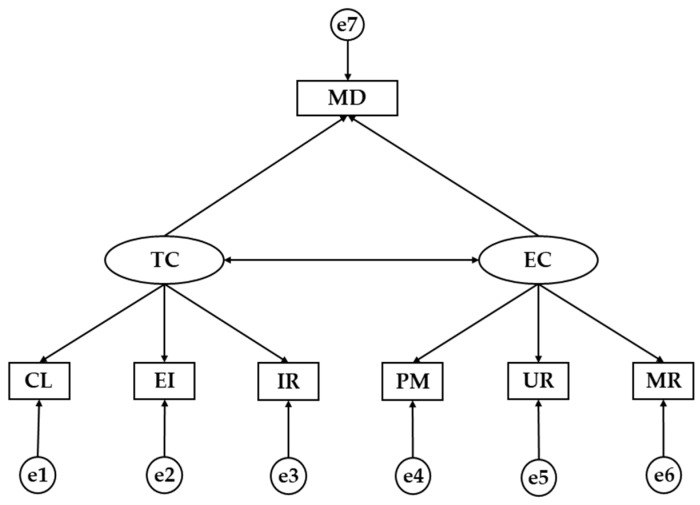
Theoretical model. TC, Task Climate; EC, Ego Climate; CL, Cooperative Learning; EI, Effort/Improvement; IR, Important Role; PM, Punishment for Mistakes; UR, Unequal Recognition; MR, Member Rivalry; MD, Mediterranean Diet.

**Figure 2 nutrients-11-00030-f002:**
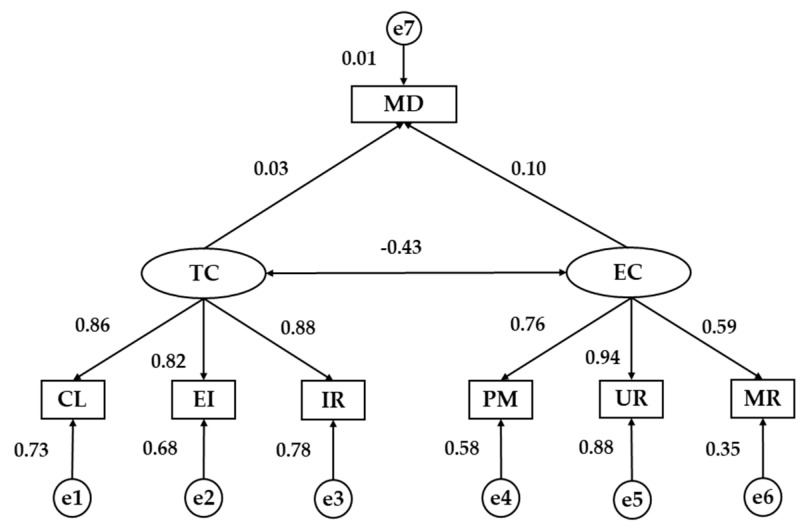
Structural Equation Model for Spanish students. TC, Task Climate; EC, Ego Climate; CL, Cooperative Learning; EI, Effort/Improvement; IR, Important Role; PM, Punishment for Mistakes; UR, Unequal Recognition; MR, Member Rivalry; MD, Mediterranean Diet.

**Figure 3 nutrients-11-00030-f003:**
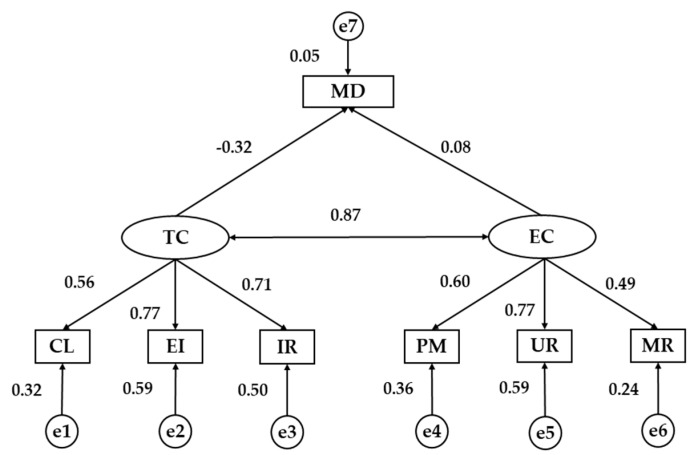
Structural Equation Model for Romanian students. TC, Task Climate; EC, Ego Climate; CL, Cooperative Learning; EI, Effort/Improvement; IR, Important Role; PM, Punishment for Mistakes; UR, Unequal Recognition; MR, Member Rivalry; MD, Mediterranean Diet.

**Table 1 nutrients-11-00030-t001:** Dietary patterns depending on country.

Items of Adherence to MD				Levene’s Test		T-Test
	Country	A	SD	F	Sig.	Sig.
I1. To eat a fruit or fruit juice every day	Romania	0.27	0.44	4.65	0.031	***
	Spain	0.77	0.41			
I2. To have a second fruit every day	Romania	0.78	0.41	119.95	0.000	***
	Spain	0.42	0.49			
I3. To eat fresh or cooked vegetables regularly once a day	Romania	0.55	0.49	6.91	0.009	0.109
	Spain	0.62	0.48			
I4. To consume fresh or cooked vegetables more than once a day	Romania	0.88	0.33	181.53	0.000	***
	Spain	0.33	0.47			
I5. Regular fish consumption (at least 2–3/week)	Romania	0.42	0.49	14.27	0.000	***
	Spain	0.67	0.46			
I6. To go ˃ 1/ week to a fast food restaurant (hamburger)	Romania	−0.56	0.49	14.83	0.000	***
	Spain	−0.34	0.47			
I7. Pulses ˃ 1/week	Romania	0.63	0.48	13.77	0.000	*
	Spain	0.71	0.45			
I8. To eat pasta or rice almost every day (5 or more per week)	Romania	0.76	0.42	106.34	0.000	***
	Spain	0.42	0.49			
I9. To have cereals or grains (bread, etc.) for breakfast	Romania	0.71	0.45	66.35	0.000	***
	Spain	0.86	0.34			
I10. To consume nuts regularly (at least 2–3/week)	Romania	0.93	0.25	1266.61	0.000	***
	Spain	0.50	0.50			
I11. To use olive oil at home	Romania	0.99	0.07	0.86	0.353	0.643
	Spain	0.99	0.09			
I12. No breakfast	Romania	−0.46	0.49	24.63	0.000	**
	Spain	−0.31	0.46			
I13. To have a dairy product for breakfast (yoghurt, milk, etc.)	Romania	0.26	0.44	15.15	0.000	***
	Spain	0.81	0.39			
I14. To eat commercially baked goods or pastries for breakfast	Romania	−0.89	0.31	0.42	0.513	***
	Spain	−0.12	0.32			
I15. To consume two yoghurts and/or some cheese (40 g) daily	Romania	0.56	0.49	2.42	0.120	0.070
	Spain	0.48	0.50			
I16. To have sweets and candy several times every day	Romania	−0.79	0.40	5.64	0.018	***
	Spain	−0.17	0.37			
Global score in adherence to MD	Rumanía	5.06	1.31	101.88	0.000	***
	España	6.65	2.63			

* Statistically significant differences at level *p* ˂ 0.05; ** Statistically significant differences at level *p* ˂ 0.01; *** Statistically significant differences at level *p* ˂ 0.001. A, Average; SD, Standard Deviation. MD, Mediterranean Diet.

**Table 2 nutrients-11-00030-t002:** Motivational climate in sport depending on country.

Dimensions of Motivational Climate	Country	A	SD	Levene’s Test		T-Test
				F	Sig.	Sig.
TC	Romania	3.11	0.55	2.720	0.100	***
	Spain	4.03	0.62			
CL	Romania	2.93	0.75	4.363	0.037	***
	Spain	4.06	0.72			
EI	Romania	3.34	0.53	4.477	0.035	***
	Spain	3.97	0.63			
IR	Romania	2.87	0.86	8.223	0.004	***
	Spain	4.10	0.73			
EC	Romania	3.24	0.54	18.175	0.000	***
	Spain	2.07	0.75			
PM	Romania	3.24	0.61	14.471	0.000	***
	Spain	1.93	0.79			
UR	Romania	3.17	0.70	6.259	0.013	***
	Spain	1.98	0.89			
MR	Romania	3.39	0.80	14.872	0.000	***
	Spain	2.56	0.95			

*** Statistically significant differences at level *p* ˂ 0.001. TC, Task-oriented Climate; CL, Cooperative Learning; EI, Effort/Improvement; IR, Important Role; EC, Ego-oriented Climate; PM, Punishment for Mistakes; UR, Unequal Recognition; MR, Member Rivalry. A, Average; SD, Standard Deviation.

**Table 3 nutrients-11-00030-t003:** Adherence to MD according to motivational climate and country.

Level of Adherence to MD According to Motivational Climate		Romania			Spain		
		A	DT	P	A	DT	P
TC	Low adherence	-	-	0.263	4.49	0.37	*
	Medium adherence	3.12	0.54		3.98	0.62	
	High adherence	2.98	0.56		4.06	0.62	
CL	Low adherence	-	-	0.677	4.60	0.37	*
	Medium adherence	2.94	0.73		4.01	0.72	
	High adherence	2.87	0.90		4.10	0.71	
EI	Low adherence	-	-	0.313	4.37	0.53	*
	Medium adherence	3.35	0.52		3.90	0.65	
	High adherence	3.23	0.61		4.02	0.60	
IR	Low adherence	-	-	0.253	4.60	0.37	0.305
	Medium adherence	2.90	0.88		4.10	0.71	
	High adherence	2.68	0.67		4.08	0.76	
EC	Low adherence	-	-	*	1.87	0.81	0.491
	Medium adherence	3.27	0.53		2.04	0.71	
	High adherence	3.00	0.54		2.11	0.78	
PM	Low adherence	-	-	*	1.66	0.39	0.391
	Medium adherence	3.28	0.61		1.89	0.75	
	High adherence	2.96	0.50		1.98	0.82	
UR	Low adherence	-	-	0.400	1.85	1.16	0.885
	Medium adherence	3.19	0.71		1.97	0.86	
	High adherence	3.05	0.70		2.00	0.92	
MR	Low adherence	-	-	**	2.33	1.17	0.225
	Medium adherence	3.45	0.76		2.49	0.90	
	High adherence	2.93	0.96		2.64	0.99	

* Statistically significant differences at level *p* ˂ 0.05; ** Statistically significant differences at level *p* ˂ 0.01. TC, Task-oriented Climate; CL, Cooperative Learning; EI, Effort/Improvement; IR, Important Role; EC, Ego-oriented Climate; PM, Punishment for Mistakes; UR, Unequal Recognition; MR, Member Rivalry. MD, Mediterranean Diet. A, Average; SD, Standard Deviation.

**Table 4 nutrients-11-00030-t004:** Structural equation model: multi-group analysis according to country.

Relationship between Variables	R.W.	S.R.W.
Estimations	S.E.	C.R.	Estimations
Weights and standardized regression weights of students from Spain
CL	←	TC	1.000	-	-	0.856 ***
EI	←	TC	0.848	0.042	19.996	0.823 ***
IR	←	TC	1.057	0.049	21.424	0.884 ***
PM	←	EC	1.000	-	-	0.759 ***
UR	←	EC	1.403	0.096	14.656	0.940 ***
MR	←	EC	0.936	0.078	12.016	0.589 ***
MD	←	TC	0.115	0.247	0.465	0.027
MD	←	EC	0.451	0.252	2.786	0.103 *
EC	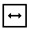	TC	-0.159	0.024	-6.723	-0.429 ***
Weights and standardized regression weights of students from Romania
CL	←	TC	1.000	-	-	0.562 ***
EI	←	TC	0.959	0.133	7.219	0.765 ***
IR	←	TC	1.434	0.208	6.905	0.708 ***
PM	←	EC	1.000	-	-	0.601 ***
UR	←	EC	1.479	0.188	7.883	0.765 ***
MR	←	EC	1.085	0.191	5.669	0.494 ***
MD	←	TC	-0.977	1.337	-3.731	-0.315 ***
MD	←	EC	0.294	1.531	0.192	0.082
EC	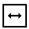	TC	0.167	0.032	5.286	0.872 ***

R.W., Regression Weights; S.R.W., Standardized Regression Weights; S.E., Estimation of Error; C.R., Critical Ratio. TC, Task Climate; CL, Cooperative Learning; EI, Effort/Improvement; IR, Important Role; EC, Ego Climate; PM, Punishment for Mistakes; UR, Unequal Recognition; MR, Member Rivalry; MD, Mediterranean Diet. * *p* ˂ 0.05; *** *p* ˂ 0.001. ←, Relationships between latent and observable indicators; 
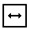
, Relationships between exogenous variables.

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
