# Peer review of "Mediterranean Diet and Motivation in Sport: A Comparative Study Between University Students from Spain and Romania"

_nutrients, 2018, doi:10.3390/nu11010030_

Round 1

Reviewer 1 Report

- Introduction section: report the role of lifestyle chanes, as a healthy diet (MD) and physical activity, to treat NAFLD, as a reported in the EASL-EASD-EASO clinical practise guidelines J Hepatol 2016. About on MD is also necessary to report the beneficial effects on health, as a weight, diabetes fatty liver (Abenavoli et al. Nutrients 2017)

- Results section: is possible to report data on the type of the sport trained by the students? E.g. run, football, swim, bike? Is possible to report data on the faculties of the students? The number of meals is the same in the groups?

- Discussion section: i suggest also to include a socio-economic analysis. Not only the climate but also the economic availability, can influence the choise of the food.

Author Response

We would like to express our gratitude for the time taken to review this manuscript and for the comments made, which we believe to be critical for producing rigorous and quality research. We have detailed below the changes made in the original article: “Mediterranean Diet and motivation in sport: A comparative study between university students from Spain and Romania” (nutrients-396476).

Modifications have been made in the original manuscript following the reviewers’ comments. For each modification we have written: the original comment as written by the reviewer in addition to the page and line number; and the change made in response to that comment. Changes have been made using the tool “Track changes” enabling editor and reviewers to identify modifications easily.

Comment 1:

Introduction section: report the role of lifestyle chanes, as a healthy diet (MD) and physical activity, to treat NAFLD, as a reported in the EASL-EASD-EASO clinical practise guidelines J Hepatol 2016. About on MD is also necessary to report the beneficial effects on health, as a weight, diabetes fatty liver (Abenavoli et al. Nutrients 2017)

Response 1:

Thank you for these indications, as they will improve the quality of the manuscript. The suggestions of reviewer 1 have been added on lines 54-63.

Comment 2:

Results section: is possible to report data on the type of the sport trained by the students? E.g. run, football, swim, bike? Is possible to report data on the faculties of the students? The number of meals is the same in the groups?

Response 2:

Thank you very much for your indications.

Unfortunately, only the KIDMED and PMCSQ-2 instruments were applied, so we do not have information about the sport practiced and the number of meals. However, we believe that this information can be very interesting for future research. Therefore, we will take them into account in future work in order to contribute new knowledge to the study area.

On the other hand, the information of the faculties from which the results were taken -(lines 107-108) has been added.

Comment 3:

Discussion section: i suggest also to include a socio-economic analysis. Not only the climate but also the economic availability, can influence the choise of the food.

Response 3:

We appreciate this suggestion.

In relation to this modification, we indicate that this information can not be included since this information was not included in the questionnaires. However, it has been included in the limitations of the study (lines).

Reviewer 2 Report

The authors have selected to study an interesting topic aiming to examine relationship between adherence to Mediterranean diet and motivation in sport in a very targeted population of University students enrolled in Physical Education degree programs in the countries of Spain and Romania.

Comments:

Major:

1. The authors mentioned that this was a cross-sectional study but similar conclusions could have been derived with an ecological study design and presenting correlations between the study variables. I believe that authors must have collected demographic and other relevant data, which could have been incorporated in the analyses to strengthen the methods and results. While a MD is considered as exposure here, there is no way to assess the influence of important factors like socioeconomic status, access to foods and other lifestyle behaviors. There is a strong precedent to conduct multiple regression analyses to derive better conclusions from this study.

6. The authors did not mention whether they were able to assess any difference in characteristics among those who were eligible but did not complete the required information to participate in the study compared to the participants.

2. Analysis these data with ANOVA in Table 3 only shows whether the motivation in any category of adherence to MD differs from the other. A regression analysis would facilitate adjusting for covariates as well as allow tests for linear trend which would elucidate the relationship between MD and motivation in sport.

3. The manuscript needs extensive editing for spelling errors, English language and style.

Minor:

1. Introduction can be edited to be more precise and should build up to the objectives without much digression.

2. In the methods, the authors mentioned that students aged 18-24 were eligible but while mentioning limitations, they state that students aged between 18-21 were analyzed. While the mean age was 20.7 (SD: 2.4), an explanation is necessary for these discrepancies.

3. A table describing the basic characteristics of the population would go a long way in addressing some issues in the manuscript. If it was published earlier, it was not referenced and a brief summary of the findings was not given.

4. The tables 2 and 3 do not define the measurement terms 'A' (average?), 'M' (Motivation score?) and 'DT'. For an audience not familiar with these measures and scales, this information would be important to understand the results.

5. The authors conclude that Task-oriented variables were associated with adherence to MD in Spain but not in Romania. This could be due to the differences variability in diet among Spaniards and Romanians and not necessarily provide meaningful conclusions for comparisons. This is further reason to consider multiple regression analyses adjusting for important covariates.

Author Response

REVIEWER  2

We would like to express our gratitude for the time taken to review this manuscript and for the comments made, which we believe to be critical for producing rigorous and quality research. We have detailed below the changes made in the original article: “Mediterranean Diet and motivation in sport: A comparative study between university students from Spain and Romania” (nutrients-396476).

Modifications have been made in the original manuscript following the reviewers’ comments. For each modification we have written: the original comment as written by the reviewer in addition to the page and line number; and the change made in response to that comment. Changes have been made using the tool “Track changes” enabling editor and reviewers to identify modifications easily.

Comment 1:

The authors have selected to study an interesting topic aiming to examine relationship between adherence to Mediterranean diet and motivation in sport in a very targeted population of University students enrolled in Physical Education degree programs in the countries of Spain and Romania.

Major:

1. The authors mentioned that this was a cross-sectional study but similar conclusions could have been derived with an ecological study design and presenting correlations between the study variables. I believe that authors must have collected demographic and other relevant data, which could have been incorporated in the analyses to strengthen the methods and results. While a MD is considered as exposure here, there is no way to assess the influence of important factors like socioeconomic status, access to foods and other lifestyle behaviors. There is a strong precedent to conduct multiple regression analyses to derive better conclusions from this study.

Response 1:

Dear reviewer,

Thank you very much for the suggestions made and the time spent in making such a thorough review.

We know the interest of including social and contextual aspects in relation to MD. Nevertheless, we regret to inform you that these data were not collected at the time the questionnaires were applied. This is because the research objective was to know the relationship between adherence to MD and motivation in sports according to country, which was approved by the ethics committee.

In this sense, we appreciate these indications and we will consider them in future research as we think that they could give more solidity to our results.

Comment 2:

The authors did not mention whether they were able to assess any difference in characteristics among those who were eligible but did not complete the required information to participate in the study compared to the participants.

Response 2:

Dear reviewer,

In this case, no assessment was done on the subjects who were not eligible, since they were discarded directly because they did not meet the inclusion criteria (so the questionnaires were not applied to them).

Comment 3:

2. Analysis these data with ANOVA in Table 3 only shows whether the motivation in any category of adherence to MD differs from the other. A regression analysis would facilitate adjusting for covariates as well as allow tests for linear trend which would elucidate the relationship between MD and motivation in sport.

Response 3:

Thanks for this indication as we think it improves the quality of the manuscript. We proceeded to perform a linear regression analysis. However, the adjustment indexes have not been adequate.

Later, we tried to make a structural equation model with multigroup analysis in relation to the country of residence (Spain or Romania). In this case, we have had good fit indexes (CFI, NFI, IFI, and RMSEA). Thus, we have decided to include this analysis since it provides the regression weights in the relationships between variables.

Comment 4:

3. The manuscript needs extensive editing for spelling errors, English language and style.

Response 4:

Thanks for this indication. The written text has been revised in order to improve it and eliminate existing errata.

Comment 5:

Minor:

1. Introduction can be edited to be more precise and should build up to the objectives without much digression.

Response 5:

Thank you very much for this indication. The introduction has been modified following the instructions of reviewer 1 and 2.

On the other hand, the authors believe that the objectives are clear and they are related to the statistical analyses performed:

1.      to assess the level of adherence to MD in a sample of university students from Spain and Romania;

2.      to analyse the relationship between adherence to MD and motivational climate in sport depending on the country (Spain or Romania)

Comment 6:

2. In the methods, the authors mentioned that students aged 18-24 were eligible but while mentioning limitations, they state that students aged between 18-21 were analyzed. While the mean age was 20.7 (SD: 2.4), an explanation is necessary for these discrepancies.

Response 6:

Thanks for this indication. This is a typo error and it has been corrected in line 244.

Comment 7:

3. A table describing the basic characteristics of the population would go a long way in addressing some issues in the manuscript. If it was published earlier, it was not referenced and a brief summary of the findings was not given.

Response 7:

Thank you very much for this indication, because we believe that its realization will bring quality to the manuscript.

The authors indicate that this table with the characteristics of the sample has not been included since these data were not available. The questionnaires that were applied only collected information about gender, age and country of residence (as well as the detailed instruments –KIDMED and PMCSQ-2), data that has already been included in the description of the sample. If the reviewer wants that we will include them as table, please let us know in order to make this change.

Comment 8:

4. The tables 2 and 3 do not define the measurement terms 'A' (average?), 'M' (Motivation score?) and 'DT'. For an audience not familiar with these measures and scales, this information would be important to understand the results.

Response 8:

Thanks for this indication. The significances of this abbreviations have been included un footnotes in both tables.

Comment 9:

5. The authors conclude that Task-oriented variables were associated with adherence to MD in Spain but not in Romania. This could be due to the differences variability in diet among Spaniards and Romanians and not necessarily provide meaningful conclusions for comparisons. This is further reason to consider multiple regression analyses adjusting for important covariates.

Response 9:

Thanks for this indication. Although we have not been able to include the regression analysis given the low indexes of adjustment of the model, we have included the conclusions obtained thanks to the structural equation model made through multi-group analysis. We hope that this can meet the requirements of the reviewer. However, if any new changes are required, do not hesitate to tell us.

Reviewer 3 Report

The aim of the present study was to evaluate the adherence to MD and its relationships with motivational climate in sport in a sample of university students from Spain and Romania. Authors suggested that it was showed a high adherence to the MD in the students from Spain. In addition, it has been demonstrated that ego-oriented climates are linked to a better adherence to MD, especially due to the importance of following a proper diet in sport contexts, as demonstrated by young Romanians. This is interesting. However, English language are check required.

Author Response

We would like to express our gratitude for the time taken to review this manuscript and for the comments made, which we believe to be critical for producing rigorous and quality research. We have detailed below the changes made in the original article: “Mediterranean Diet and motivation in sport: A comparative study between university students from Spain and Romania” (nutrients-396476).

Modifications have been made in the original manuscript following the reviewers’ comments. For each modification we have written: the original comment as written by the reviewer in addition to the page and line number; and the change made in response to that comment. Changes have been made using the tool “Track changes” enabling editor and reviewers to identify modifications easily.

Comment 1:

The aim of the present study was to evaluate the adherence to MD and its relationships with motivational climate in sport in a sample of university students from Spain and Romania. Authors suggested that it was showed a high adherence to the MD in the students from Spain. In addition, it has been demonstrated that ego-oriented climates are linked to a better adherence to MD, especially due to the importance of following a proper diet in sport contexts, as demonstrated by young Romanians. This is interesting. However, English language are check required.

Response 1:

Thanks for all your indications as they will improve the quality of the manuscript. The English text has been revised and corrected by a native English speaker.

Round 2

Reviewer 2 Report

Dear Authors,

Thank you for taking the time and efforts to make considerable changes to this manuscript. As in my previous review report, this research has great potential but lacks some key elements to derive strong conclusions. However, the authors have clearly acknowledged the shortcomings of the dataset.

I have the following suggested changes in the manuscript. The text in red indicates suggested changes or edits:

Line 38: Being overweight (body mass index of 30 pr higher - please indicate appropriately if it is defined differently for the study regions) and its associated pathologies have become a problem in society nowadays, which is why the importance of following a healthy diet and lifestyle has gained much prominence.

Line 43: Utilizing different dietary models, several authors state the importance of making a change in people's eating habits in European countries, in order to....

Line 45: Thus, the Mediterranean Diet

Line 48: risk of suffer diabetes, bone diseases and overweight gaining excess weight.

Line 56: socio-economic contexts

Line 61: helps to prevent diabetes... 

Line 68: Regular and adequate The practice of Physical Activity has become in addition to following a healthy diet is an essential for the improvement of health.

Line 72: ...... PA decreases anxiety and stress and improves self concept....

Line 75: in early adulthood adulthood are reinforced....

Line 80: explaining the ...

Line 82: by each person an individual...

Line 97:  in  order to discover the keys for to promote this  dietary modelMD in...

Line 100: parameters on in a sample...

Line 110: enrolled onin Physical Education degrees,....

Line 13: strongly agree

Line 142: during normalregular classes...

Line 146: received any incentives.

Line 149: analysis was developed conducted using..

Line 156: USA) iswas used...

Line 182: ...Physical Education degrees.

Line 186: Spanish students obtained higher scores

Line 280: scores - in what?

Line 280: However, these higher scores...

Line 281: consumption in among Romaninan....  Please rewrite the sentence in Line 281.

Line 337: Suggested edits: The present study shows evidence for a higher adherence to MD among Spanish students, as indicated by a higher intake of fruits, vegetables, cereals and fish.

Line 340: The term "junk" was not defined anywhere. I suggest to state is a "high cholesterol and carbohydrate" or "fried foods" or other appropriate terms.

Line 343: determined suggested

Line 351: Suggested edits: .....not related to nutrition. While further research is necessary to substantiate the findings from this study, there are adequate indications to pursue development of intervention programs.....

Please go through the manuscript again carefully to improve language structure and style. Please also correct excessive spacing between words in some paragraphs (e.g. Line 183)

Author Response

REVIEWER 2
Dear reviewer,
We would like to express our gratitude for the time taken to review this manuscript and for the comments made, which we believe to be critical for producing rigorous and quality research. We have detailed below the changes made in the original article: “Mediterranean Diet and motivation in sport: A comparative study between university students from Spain and Romania” (nutrients-396476).
Modifications have been made in the original manuscript. For each modification we have written: the original comment as written by the reviewer in addition to the page and line number; and the change made in response to that comment. Changes have been made using the tool “Track changes” enabling editor and reviewers to identify modifications easily.
Comments:

Dear Authors,

Thank you for taking the time and efforts to make considerable changes to this manuscript. As in my previous review report, this research has great potential but lacks some key elements to derive strong conclusions. However, the authors have clearly acknowledged the shortcomings of the dataset.

I have the following suggested changes in the manuscript. The text in red indicates suggested changes or edits:

Line 38: Being overweight (body mass index of 30 pr higher - please indicate appropriately if it is defined differently for the study regions) and its associated pathologies have become a problem in society nowadays, which is why the importance of following a healthy diet and lifestyle has gained much prominence.

Line 43: Utilizing different dietary models, several authors state the importance of making a change in people's eating habits in European countries, in order to....

Line 45: Thus, the Mediterranean Diet

Line 48: risk of suffer diabetes, bone diseases and overweight gaining excess weight.

Line 56: socio-economic contexts

Line 61: helps to prevent diabetes... 

Line 68: Regular and adequate The practice of Physical Activity has become in addition to following a healthy diet is an essential for the improvement of health.

Line 72: ...... PA decreases anxiety and stress and improves self concept....

Line 75: in early adulthood adulthood are reinforced....

Line 80: explaining the ...

Line 82: by each person an individual...

Line 97:  in  order to discover the keys for to promote this  dietary modelMD in...

Line 100: parameters on in a sample...

Line 110: enrolled onin Physical Education degrees,....

Line 13: strongly agree

Line 142: during normalregular classes...

Line 146: received any incentives.

Line 149: analysis was developed conducted using..

Line 156: USA) iswas used...

Line 182: ...Physical Education degrees.

Line 186: Spanish students obtained higher scores

Line 280: scores - in what?

Line 280: However, these higher scores...

Line 281: consumption in among Romaninan....  Please rewrite the sentence in Line 281.

Line 337: Suggested edits: The present study shows evidence for a higher adherence to MD among Spanish students, as indicated by a higher intake of fruits, vegetables, cereals and fish.

Line 340: The term "junk" was not defined anywhere. I suggest to state is a "high cholesterol and carbohydrate" or "fried foods" or other appropriate terms.

Line 343: determined suggested

Line 351: Suggested edits: .....not related to nutrition. While further research is necessary to substantiate the findings from this study, there are adequate indications to pursue development of intervention programs.....

Response 1:
Thank you very much for the time spent in reviewing the manuscript, since we estimate that the reviewer has made a great effort to read the article in such detail.
All the writing instructions that the reviewer exhibits have been made. We would like to personally thank the implication of him, since without any doubt these changes will improve the quality of the article.
Comment 2:
Please go through the manuscript again carefully to improve language structure and style. Please also correct excessive spacing between words in some paragraphs (e.g. Line 183)
Response 2:
Thank you very much for this indication. We have reviewed the manuscript and deleted the unnecessary spaces.
We remain at your entire disposal for any new change that you deem appropriate.
Sincerely,
The authors.